# Hospital Readmission Risk and Risk Factors of People with a Primary or Secondary Discharge Diagnosis of Diabetes

**DOI:** 10.3390/jcm12041274

**Published:** 2023-02-06

**Authors:** Daniel J. Rubin, Naveen Maliakkal, Huaqing Zhao, Eli E. Miller

**Affiliations:** 1Section of Endocrinology, Diabetes, and Metabolism, Lewis Katz School of Medicine, Temple University, 3322 N. Broad Street, Suite 205, Philadelphia, PA 19140, USA; 2Department of Medicine, Temple University Hospital, Philadelphia, PA 19140, USA; 3Department of Biomedical Education and Data Science, Lewis Katz School of Medicine, Temple University, 3322 N. Broad Street, Suite 205, Philadelphia, PA 19140, USA

**Keywords:** diabetes, readmission, risk factors

## Abstract

Hospital readmission among people with diabetes is common and costly. A better understanding of the differences between people requiring hospitalization primarily for diabetes (primary discharge diagnosis, 1°DCDx) or another condition (secondary discharge diagnosis, 2°DCDx) may translate into more effective ways to prevent readmissions. This retrospective cohort study compared readmission risk and risk factors between 8054 hospitalized adults with a 1°DCDx or 2°DCDx. The primary outcome was all-cause hospital readmission within 30 days of discharge. The readmission rate was higher in patients with a 1°DCDx than in patients with a 2°DCDx (22.2% vs. 16.2%, *p* < 0.01). Several independent risk factors for readmission were common to both groups including outpatient follow up, length of stay, employment status, anemia, and lack of insurance. C-statistics for the multivariable models of readmission were not significantly different (0.837 vs. 0.822, *p* = 0.15). Readmission risk of people with a 1°DCDx was higher than that of people with a 2°DCDx of diabetes. Some risk factors were shared between the two groups, while others were unique. Inpatient diabetes consultation may be more effective at lowering readmission risk among people with a 1°DCDx. These models may perform well to predict readmission risk.

## 1. Introduction

Readmission to the hospital is an undesirable outcome. Thus, there is widespread interest in reducing readmission risk to improve both the patient health and control costs [1,2,3]. It has been established that diabetes is an independent risk factor for readmission [4,5]. Furthermore, the sheer number of readmissions and their associated costs among people with diabetes are staggering. In the U.S., there were more than eight million hospital discharges of patients with diabetes, accounting for nearly 30% of all discharges in 2018 [6,7]. At that time, 10.5% of the U.S. population had diabetes, a difference that reflects the overall hospitalization risk associated with diabetes [8]. Given the 16.0 to 20.4% rate of readmission within 30 days of discharge (30-day readmission) [9,10], the annual cost of such readmissions is $20–25 billion in the U.S. alone. Of note, most hospitalized patients with diabetes have type 2 diabetes, reflecting the underlying prevalence of type 2 diabetes in the general population [7].

Over the past several years, there have been multiple efforts to determine the risk factors for readmission among patients with diabetes [9,10]. Many risk factors across several domains have been identified including sociodemographics, diabetic complications, comorbidity burden, abnormal laboratory values, multiple hospitalizations, and hospital length of stay. Patients with diabetes, however, are a heterogeneous population that can be categorized as requiring hospitalization primarily for diabetes (primary discharge diagnosis) or another condition (secondary discharge diagnosis of diabetes). One study found that hospitalized patients with a primary diagnosis of diabetes had a higher risk of readmission than patients with a secondary diagnosis of diabetes [11], suggesting that the readmission risk factors of these two populations may be different. Whether or not the risk factors for readmission vary by the primary or secondary discharge diagnosis of diabetes is unknown. A better understanding of the differences between these populations may translate into more effective ways to prevent readmissions.

To compare the readmission risk and risk factors between patients with a primary or secondary discharge diagnosis of diabetes, we performed a secondary analysis of a previously described cohort [12].

## 2. Methods

### 2.1. Study Sample

This retrospective cohort study was based on electronic medical records of 17,284 patients with 44,203 hospital discharges between 1 January 2004 and 1 December 2012 at Boston Medical Center, an urban academic medical center in Boston, MA, as previously described [12]. The inclusion criteria were a diagnosis of diabetes defined by a hospital discharge associated with an International Classification of Diseases, Ninth Revision, Clinical Modification (ICD-9-CM) code of 250.xx or preadmission documentation of a diabetes medication. Patients were excluded for the following: age less than 18 years on the day of an admission, discharge by transfer to another hospital, discharge from an obstetric service, inpatient death, outpatient death within 30 days of discharge, missing data, or lack of follow-up 30 days after discharge. A readmission within 8 h after an index discharge was considered as a false positive and merged with the index discharge to avoid counting an in-hospital transfer as a readmission. Among the discharges with a primary diagnosis of diabetes, simple random sampling was used to select one discharge per patient, without replacement, yielding 4027 discharges. Among the patients with only secondary discharge diagnoses of diabetes, 4027 discharges were randomly selected, one discharge per patient. A post-hoc analysis was performed in the subgroup of 3674 patients who had an HbA1c value available. The Temple University Institutional Review Board approved the protocol.

### 2.2. Definition of Variables and Outcomes

A total of 49 sociodemographic, clinical, and administrative variables linked with hospital discharges were evaluated for their association with all-cause hospital readmission within 30 days of discharge, as previously described [12]. The first value of each variable up to 24 h before the admission was analyzed so that the related outpatient and emergency department visits were included. The most extreme blood glucose level was based on capillary point-of-care or venous values during the entire hospitalization. The most extreme value was placed into one of three categories: 70–180 mg/dL (3.9–10 mmol/L), 40–69 or 181–300 mg/dL (2.2–3.8 or 10.1–16.7 mmol/L), or <40 or >300 mg/dL (<2.2 or >16.7 mmol/L). The most common ICD-9-CM codes within each cohort were grouped by condition or organ system and sorted by frequency. Inpatient consultation by the diabetes management team was assessed as present or absent. These consultations were requested by primary hospital providers. The team consisted of a nurse practitioner/certified diabetes educator, an endocrinology fellow, and an endocrinology attending with expertise in diabetes. Consultations may have consisted of a single visit or intermittent or daily follow-up visits with co-management throughout the hospital stay including recommendations for diabetes management upon discharge.

### 2.3. Statistical Analyses

Summaries of the categorical variables included the counts and percentages. For continuous variables, the means and standard deviations or medians and interquartile ranges were used accordingly after the assessment of normality. The characteristics of patients with a primary diagnosis of diabetes were compared to the characteristics of those with a secondary diagnosis of diabetes. In addition, readmitted and non-readmitted patients were compared among those with a primary diagnosis of diabetes and among those with a secondary diagnosis of diabetes. For the categorical variables, these comparisons were conducted by χ^2^ tests. For continuous variables, 2-sample t tests or Wilcoxon rank sum tests were used. For multivariable modeling, non-normally distributed continuous variables (admission serum creatinine and length of stay) were log transformed. Univariate analyses identified variables associated with 30-day readmission. Variables with *p* < 0.1 in the univariate analyses were selected to undergo multivariable modeling. To determine the adjusted associations of the variables with all-cause 30-day readmission, multivariable logistic regression with generalized estimating equations and the best subset selection was performed [13,14]. The threshold for retention in the multivariable models was an association with 30-day all-cause readmission at *p* < 0.05. A *p*-value < 0.05 was considered statistically significant. To explore the performance of the models for readmission prediction, c-statistics, a measure of discrimination representing the area under the receiver operating characteristic curve [15], were calculated with 95% confidence intervals and calibration plots were drawn [16]. All analyses were performed using SAS version 9.4 (SAS Institute, Cary, NC, USA).

## 3. Results

A total of 8054 patients were analyzed: 4027 with a primary discharge diagnosis of diabetes and 4027 with a secondary discharge diagnosis of diabetes (Table 1). The cohort was ethnically diverse (40.4% Black, 24.3% White, 12.5% Hispanic), well-distributed across four age brackets, and balanced for sex (47.1% female). Most of the patients were unmarried, educated at a high-school level or greater, not employed, insured by Medicare or Medicaid, and lived within 5 miles of the hospital. Nearly 40% of patients had at least one microvascular diabetic complication and almost 50% had at least one macrovascular complication. The most common comorbidities other than diabetic complications were hypertension, anemia, and depression. The median hospital length of stay was 3.3 days.

Out of the 49 characteristics analyzed, 44 were statistically significantly different between patients with a primary and secondary discharge diagnosis of diabetes (Table 1). There were no statistically significant differences in preadmission thiazolidinedione use, admission hematocrit, intensive care unit admission, a diagnosis of anemia ever, or current infection during the admission.

The readmission rate was higher in patients with a primary discharge diagnosis of diabetes than in patients with a secondary discharge diagnosis of DM (22.2% vs. 16.2%, *p* < 0.01). Several independent risk factors for readmission were common to both a primary and a secondary discharge diagnosis of diabetes, specifically, a lack of an outpatient visit within 30 days of discharge, length of stay, being unemployed, being discharged within 90 days before admission, and a diagnosis of anemia (Figure 1 and Figure 2). Being uninsured was associated with lower readmission risk. There were also multiple independent readmission risk factors unique to patients with a primary discharge diagnosis of diabetes (Figure 3): the Charlson comorbidity index, education level, gastroparesis, higher serum creatinine, and lower hematocrit. Inpatient diabetes consultation and preadmission TZD use were associated with lower odds of readmission in this group. Similarly, there were several independent readmission risk factors unique to those with a secondary discharge diagnosis of diabetes (Figure 4): discharge against medical advice, discharge home with nursing care, pancreatitis, abnormal serum sodium, urgent or emergent admission, and low serum albumin.

C-statistics for the multivariable models of readmission indicated very good discrimination and were not significantly different between the study groups (0.837 [0.823–0.851] 95% CI vs. 0.822 [0.807–0.837] 95% CI, *p* = 0.15). Calibration of the primary discharge diagnosis model was excellent, while calibration of the secondary discharge diagnosis model was fair (Figure 5 and Figure 6).

Many of the most frequent reasons for hospital admission based on primary ICD-9-CM code among patients with a secondary discharge diagnosis of diabetes were also frequent secondary ICD-9-CM codes among those with a primary discharge diagnosis of diabetes (i.e., cardiovascular disease, infection, lung disease, procedure or postoperative complications, and disorders of fluid electrolyte or acid–base balance, Appendix A). Other common reasons for admission in the patients with a secondary discharge diagnosis of diabetes were ischemic stroke, alteration of consciousness, hallucinations, syncope, convulsions, dizziness, fever, or malaise, overweight, obesity and other hyperalimentation, and pancreatitis (Appendix A).

In the subgroup analysis performed among patients with an HbA1c value, the mean HbA1c in the primary discharge diagnosis group was 10.7 ± 3.0% and in the secondary discharge diagnosis group, it was 7.8 ± 2.0% (*p* < 0.001). When HbA1c was added to the models for readmission, there was no association of HbA1c with readmission in either group of patients (Appendix A). Although two and three of the variables in each model were no longer statistically significant, the direction of the odds ratios above or below 1 remained the same.

## 4. Discussion

In this retrospective cohort study of 8054 hospitalized patients with either a primary or secondary diagnosis of diabetes, 49 socioeconomic, demographic, clinical, and administrative variables were evaluated for associations with all-cause 30-day readmission. Multivariable analysis revealed several independent risk factors for readmission, some of which were shared between the two study groups and some of which were not. Both models performed well in terms of discrimination (c-statistics 0.834 and 0.822), suggesting very good performance for prediction, with no statistically significant difference between them. Post-hoc analysis in the subgroup of patients with an HbA1c value found no association of HbA1c with readmission and did not substantively change the model in either group. The loss of statistical significance in a few of the variables in the models was attributable to the markedly smaller sample sizes. Finally, patients with a primary discharge diagnosis of diabetes had a significantly higher readmission rate than those with a secondary discharge diagnosis of diabetes.

The higher readmission rate of patients with a primary discharge diabetes diagnosis was consistent with the existing literature as well as our clinical experience. In a study of 16,266 people with diabetes, Sonmez and others reported 30-day readmission rates of 16.5% and 13.6% among those with a primary or secondary discharge diagnosis of diabetes, respectively [11]. Another study of adults with type 1 diabetes hospitalized for diabetic ketoacidosis (DKA) reported a readmission rate of 19.4% [17]. It has been speculated, and we agree, that the higher readmission rate of those with a primary discharge diagnosis of diabetes may be related to the more extreme metabolic abnormalities (e.g., diabetic ketoacidosis, hyperglycemic hyperosmolar state, severe hyper and hypoglycemia) that patients tend to have relative to those for whom diabetes is a secondary diagnosis [11]. Sonmez and colleagues called for studies to reveal the causes for the observed difference in readmission rates between the two populations. We are not aware of studies besides ours that compared the multiple risk factors between patients with a primary or secondary discharge diagnosis of diabetes.

The difference in readmission rates between patients with a primary or secondary diabetes diagnosis may be at least partly attributable to the risk factors that were unique to each subgroup. Most notably, inpatient consultation by a diabetes management service was associated with lower odds of readmission in people with a primary discharge diagnosis of diabetes. This association has been reported in other studies of hospitalized patients with diabetes including one randomized controlled trial [18,19,20,21]. These studies, however, did not distinguish between those with a primary or secondary discharge diagnosis of diabetes. The literature, our findings, and clinical intuition considered together suggest that inpatient diabetes team consultation is more effective at reducing readmission risk among patients primarily admitted for diabetes than in patients with diabetes admitted for another condition.

Risk factors unique to patients with a secondary discharge diagnosis of diabetes include being discharged against medical advice, being discharged home with nursing care, and urgent or emergent admission. In contrast to inpatient diabetes consultation, these factors are not diabetes specific. Given that diabetes is not the central issue among those with a secondary diagnosis, it is logical that non-specific risk factors are more important than among those with a primary diagnosis of diabetes. For secondary diabetes diagnosis patients, it appears that the circumstances around the hospital admission and discharge are more important for determining the readmission risk than for primary diabetes diagnosis patients.

It is worth noting that the risk factors common to both primary and secondary diabetes diagnosis patients (i.e., lack of an outpatient visit within 30 days of discharge, length of stay, being unemployed, lacking health insurance, being discharged within 90 days before admission, and a diagnosis of anemia) are also not specific to diabetes per se. In addition, the only modifiable factors on this list are outpatient follow-up and insurance status. There is some support from randomized controlled trials for the hypothesis that in-person outpatient follow-up reduces readmission risk in people with diabetes [22,23], although another trial has provided conflicting evidence [24], and the nature of follow-up and the study population characteristics vary across the few trials that have examined this. Whether follow-up by telephone and specific components of outpatient follow-up such as education and medication adjustment contribute to readmission risk reduction has been reviewed elsewhere recently and was beyond the scope of the current study [9]. Lack of health insurance was strongly and inversely associated with readmission risk in both groups of patients. Previously, we reported this association in a larger study of the parent cohort from which the current sample was drawn [12]. We speculate that patients who lack insurance delay seeking care to avoid paying medical bills. The association of hospital length of stay with readmission has been widely reported [9,10], and likely represents a marker of illness severity rather than a causal factor. It is unclear why a diagnosis of anemia is the only condition among all the diagnoses evaluated to be shared as a risk factor for readmission in both primary and secondary diabetes diagnosis patients. Additional research to both confirm and explore reasons for this association is warranted.

There is some commonality and some differences between the risk factors reported here and a study of adults with type 1 diabetes hospitalized with DKA [17]. Common risk factors were Charlson comorbidity index and kidney disease. Risk factors identified in the other study not identified in ours among patients with a primary discharge diagnosis of diabetes include age, sex, income, large hospital bed size, smoking, discharge against medical advice, obesity, and hypertension. These differences likely reflect differences in the data available as well as the study populations. The other study used a national sample of much younger patients and did not include much patient-level data such as laboratory results and medication use.

The c-statistics of the two models presented here suggest very good prediction of readmissions and compare favorably to other models that predict readmission risk in people with diabetes, for which the c-statistics ranged from 0.63 to 0.97 [9]. We previously published a model using the parent cohort of people with diabetes without stratifying the sample by discharge diagnosis, which had a comparable c-statistic of 0.82 [12]. These two studies indicate that developing separate models for patients with either a primary or secondary discharge diagnosis of diabetes did not yield better performance than a single model developed in a unified cohort. They also reinforced the conclusion that using more variables, especially variables based on data available on or after discharge, enabled stronger prediction than models based only on variables available at the time of admission such as the Diabetes Early Readmission Risk Indicator (DERRI^®^), which had a c-statistic of 0.69 [25]. It remains unknown whether models might perform better when stratified by other characteristics such as the type of diabetes.

It is difficult to speculate how the readmission risk factors among people with diabetes may have changed since the appearance of COVID-19. Given that the pandemic exacerbated health disparities [26], it is possible that the associations related to access to care such as outpatient visits, employment status, and insurance were strengthened. While several studies have identified diabetes as a risk factor for readmission among people hospitalized for COVID-19 [27], we are unaware of any studies that have examined COVID-19 as a risk factor for readmission among people with diabetes.

The strengths of this study are a moderately large sample size, a diverse population, and analysis of multiple socioeconomic, demographic, administrative, and clinical factors. These strengths are tempered by some limitations. Because the sample came from one urban academic medical center, the results may not be generalizable to other settings and populations. Additionally, readmissions that may have occurred at other hospitals could not be assessed. However, given that the readmission rate of 20.4% in the parent cohort is at the higher end of the range reported for people with diabetes [9,10], it seems unlikely that a substantial number of people were readmitted at other hospitals. Post-discharge mortality data were not available, and different mortality rates between the two groups may have influenced the observed readmission rates. Data on other potentially important risk factors or confounders such as A1c (due to lack of collection in about half the cohort), diabetes type (for which the accuracy of ICD-9-CM codes is suboptimal) [28], inpatient management, and classification of primary teams as medical or surgical were not available to analyze. Furthermore, the study period ended before FDA approval of SGLT2-inhibitors and the widespread use of GLP1-receptor agonists, which are drug classes that may influence the risk of hospitalization and readmission. Finally, the observational nature of this study precludes causal inference.

In conclusion, this retrospective observational study of patients with a primary or secondary discharge diagnosis of diabetes identified shared unique risk factors for all-cause 30-day readmission while confirming the higher readmission risk of patients with a primary diabetes diagnosis. The results suggest that inpatient diabetes consultation may be more effective at lowering the readmission risk among patients with a primary diabetes diagnosis than those with a secondary diabetes diagnosis. Given the burden incurred and imposed by hospital readmissions among people with diabetes, identifying those at greater risk of readmission offers the potential to allocate resources more efficiently and effectively to reduce readmission risk. These models may perform well to predict readmission risk, although additional study is needed to validate their performance. Randomized controlled trials are needed to test the strategy of linking readmission risk prediction with interventions for reducing such risk.

## Figures and Tables

**Figure 1 jcm-12-01274-f001:**
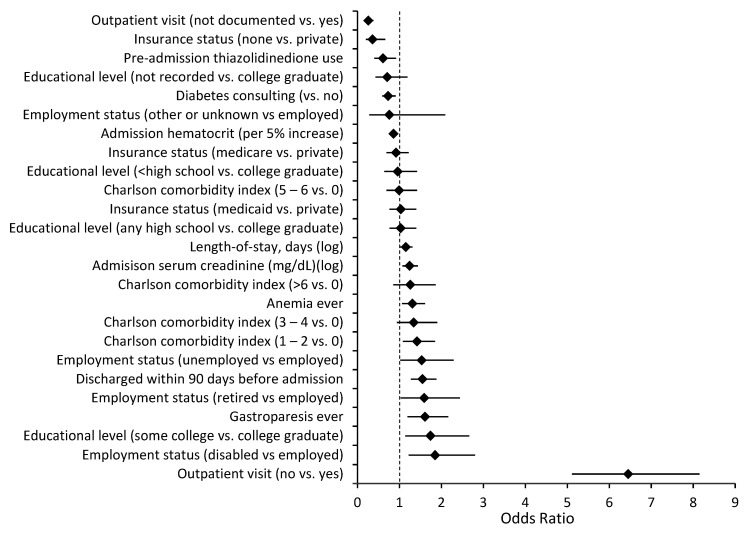
Risk factors for all-cause 30-day readmission among 4027 patients with primary discharge diagnosis of diabetes in multivariable logistic regression model, OR (95% CI), adjusted for year of discharge. ♦ = Odds ratio.

**Figure 2 jcm-12-01274-f002:**
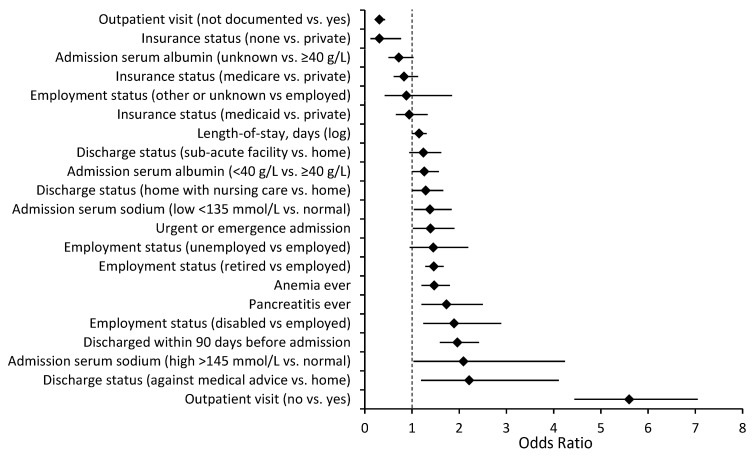
Risk factors for all-cause 30-day readmission among 4027 patients with secondary discharge diagnosis of diabetes in the multivariable logistic regression model, OR (95% CI). Adjusted for year of discharge. ♦ = Odds ratio.

**Figure 3 jcm-12-01274-f003:**
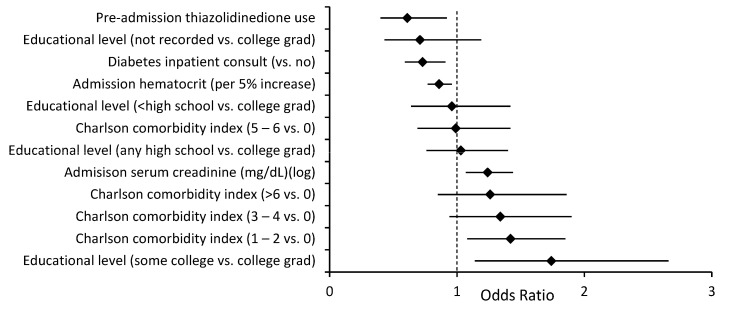
Risk factors for readmission unique to primary discharge diagnosis of diabetes, OR (95% CI). ♦ = Odds ratio.

**Figure 4 jcm-12-01274-f004:**
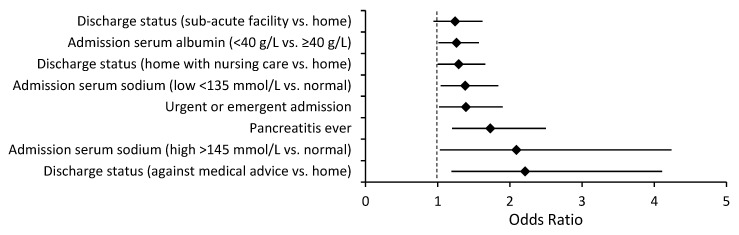
Risk factors for readmission unique to the secondary discharge diagnosis of diabetes, OR (95% CI). ♦ = Odds ratio.

**Figure 5 jcm-12-01274-f005:**
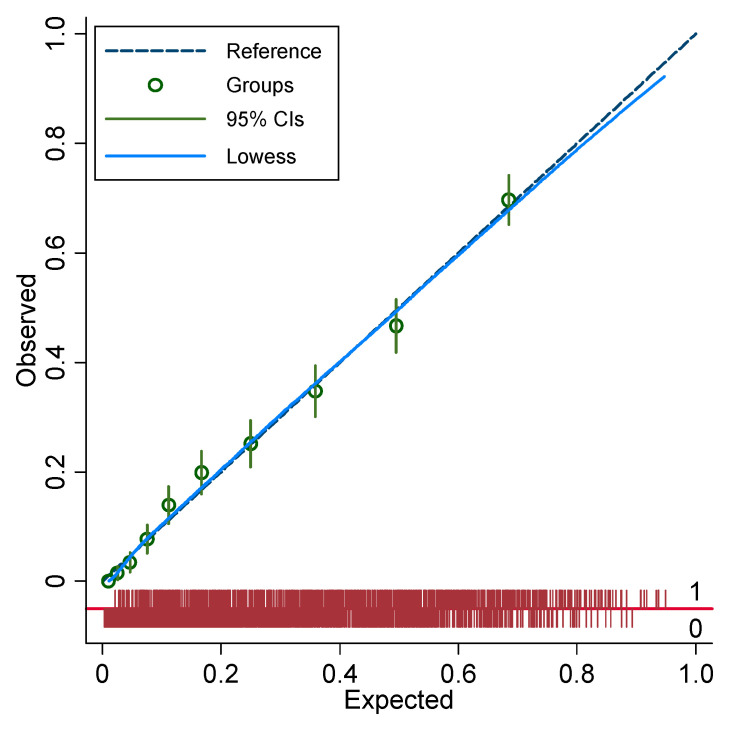
Calibration plot for the multivariable logistic regression model in people with primary discharge diagnosis of diabetes. Each decile is denoted by a circle with a short intersecting line to indicate the corresponding 95% confidence interval. The diagonal smooth line (Lowess) indicates excellent agreement between the observed and expected values.

**Figure 6 jcm-12-01274-f006:**
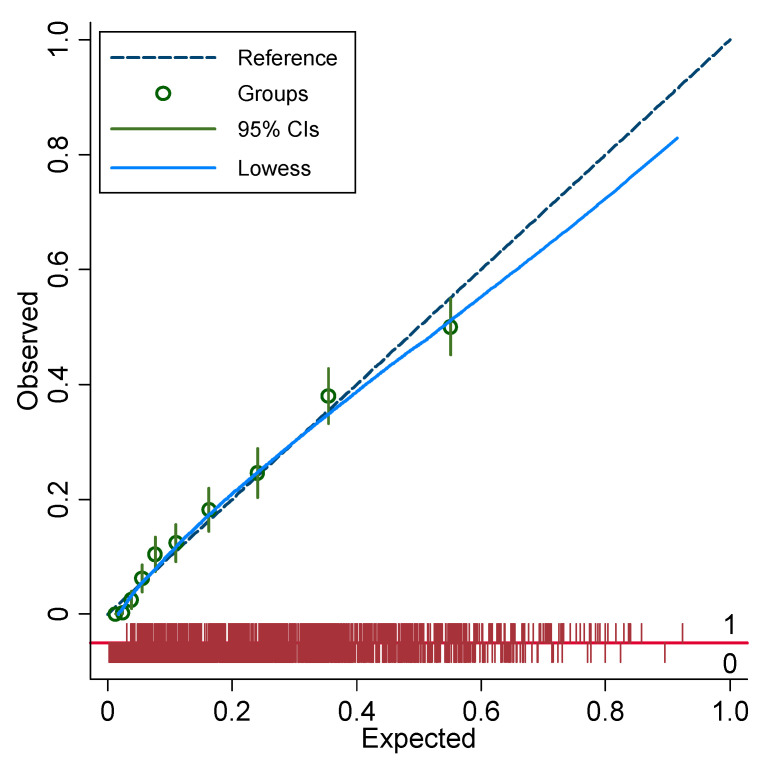
Calibration plot for the multivariable logistic regression model in people with secondary discharge diagnosis of diabetes. Each decile is denoted by a circle with a short intersecting line to indicate the corresponding 95% confidence interval. The diagonal smooth line (Lowess) indicates fair agreement between the observed and expected values.

**Table 1 jcm-12-01274-t001:** Characteristics of the hospitalized patients by the primary or secondary discharge diagnosis of diabetes.

Variable	All Patients N = 8054	Primary Diabetes Dx N = 4027	Secondary Diabetes Dx N = 4027	*p* Value
Age, N (%)				<0.0001
<50 years	2246 (27.9)	1557 (38.7)	689 (17.1)	
50–59 years	1890 (23.5)	972 (24.1)	918 (22.8)	
60–69 years	1808 (22.4)	771 (19.1)	1037 (25.8)	
70+ years	2110 (26.2)	727 (18.1)	1383 (34.3)	
Female, N (%)	3796 (47.1)	1757 (43.6)	2039 (50.6)	0.0004
Marital status, ^a^ N (%)				<0.0001
Married	2337 (29.0)	948 (23.5)	1389 (34.5)	
Single	5558 (69.0)	3018 (74.9)	2540 (63.1)	
Race/ethnicity, ^a^ N (%)				<0.0001
Black	3254 (40.4)	1990 (49.4)	1264 (31.4)	
Hispanic	1008 (12.5)	509 (12.6)	499 (12.4)	
White	1956 (24.3)	785 (19.5)	1171 (29.1)	
Not recorded	1522 (18.9)	616 (15.3)	906 (22.5)	
English speaking, N (%)	6569 (81.6)	3409 (84.7)	3160 (78.5)	<0.0001
Insurance status, N (%)				<0.0001
Medicaid	1592 (19.8)	948 (23.5)	644 (16.0)	
Medicare	2935 (36.4)	1366 (33.9)	1569 (39.0)	
None	469 (5.8)	334 (8.3)	135 (3.4)	
Private	1614 (20.0)	780 (19.4)	834 (20.7)	
Not recorded	1444 (17.9)	599 (14.9)	845 (21.0)	
Home zip code < 5 mi. from hospital, N (%)	5688 (70.6)	3121 (77.5)	2567 (63.7)	<0.0001
Educational level, N (%)				<0.0001
Less than high school	1039 (12.9)	503 (12.5)	536 (13.3)	
Any high school	4465 (55.4)	2366 (58.8)	2099 (52.1)	
Some college	535 (6.6)	294 (7.3)	241 (6.0)	
College graduate	1261 (15.7)	571 (14.2)	690 (17.1)	
Not recorded	754 (9.4)	293 (7.3)	461 (11.4)	
Employment, ^a^ N (%)				<0.0001
Disabled	1742 (21.6)	1033 (25.7)	709 (17.6)	
Employed	885 (11.0)	410 (10.2)	475 (11.8)	
Retired	2536 (31.5)	941 (23.4)	1595 (39.6)	
Unemployed	2635 (32.7)	1551 (38.5)	1084 (26.9)	
Pre-admission sulfonylurea use, N (%)	1066 (13.2)	408 (10.1)	658 (16.3)	<0.0001
Pre-admission metformin use, N (%)	2029 (25.2)	753 (18.7)	1276 (31.7)	<0.0001
Pre-admission insulin use, N (%)	3404 (42.3)	2216 (55.0)	1188 (29.5)	<0.0001
Steroids at admission	595 (7.4)	222 (5.5)	373 (9.3)	<0.0001
Most extreme blood glucose level, ^b^ N (%)				<0.0001
40–69 or 181–300 mg/dL	3008 (37.3)	1238 (30.7)	1770 (44.0)	
70–180 mg/dL	2182 (27.1)	502 (12.5)	1680 (41.7)	
<40 or >300 mg/dL	2864 (35.6)	2287 (56.8)	577 (14.3)	
Diabetes inpatient consultation, N (%)	1854 (23.0)	1438 (35.7)	416 (10.3)	<0.0001
Current or prior DKA or HHS, N (%)	1471 (18.3)	1416 (35.2)	55 (1.4)	<0.0001
Microvascular complications, ^c^ N (%)				<0.0001
0	5099 (63.3)	1938 (48.1)	3161 (78.5)	
1	1781 (22.1)	1164 (28.9)	617 (15.3)	
2	781 (9.7)	588 (14.6)	193 (4.8)	
3	393 (4.9)	337 (8.4)	56 (1.4)	
Macrovascular complications, ^d^ N (%)				<0.0001
0	4207 (52.2)	2346 (58.3)	1861 (46.2)	
1	2126 (26.4)	967 (24.0)	1159 (28.8)	
2	1228 (15.2)	447 (11.1)	781 (19.4)	
3	393 (4.9)	217 (5.4)	176 (4.4)	
4	100 (1.2)	50 (1.2)	50 (1.2)	
Pre-admission blood pressure meds, N (%)				<0.0001
None	2743 (34.1)	1521 (37.8)	1222 (30.3)	
ACE-i or ARB	3699 (45.9)	1789 (44.4)	1910 (47.4)	
Non-ACE or ARB	1612 (20.0)	717 (17.8)	895 (22.2)	
Pre-admission statin use, N (%)	3389 (42.1)	1462 (36.3)	1927 (47.9)	<0.0001
Admission white blood cell count, N (%)				<0.0001
Low < 4 K/μL	387 (4.8)	218 (5.4)	169 (4.2)	
Normal 4–11 K/μL	6278 (77.9)	3228 (80.2)	3050 (75.7)	
High > 11 K/μL	1389 (17.2)	581 (14.4)	808 (20.1)	
Admission serum albumin, N (%)				<0.0001
4+ g/dL	3116 (38.7)	1722 (42.8)	1394 (34.6)	
<4 g/dL	4088 (50.8)	1984 (49.3)	2104 (52.2)	
Unknown	850 (10.6)	321 (8.0)	529 (13.1)	
Admission serum sodium, N (%)				<0.0001
Low < 135 mmol/L	914 (11.3)	533 (13.2)	381 (9.5)	
Normal 135–145 mmol/L	7078 (87.9)	3470 (86.2)	3608 (89.6)	
High > 145 mmol/L	62 (0.8)	24 (0.6)	38 (0.9)	
Admission serum potassium, N (%)				<0.0001
Low < 3.1 mmol/L	95 (1.2)	46 (1.1)	49 (1.2)	
Normal 3.1–5.3 mmol/L	7196 (89.3)	3473 (86.2)	3723 (92.5)	
High > 5.3 mmol/L	763 (9.5)	508 (12.6)	255 (6.3)	
Admission creatinine (mg/dL), median (IQR)	0.9 (0.7–1.3)	1.0 (0.8–1.4)	0.9 (0.7–1.3)	0.0012
Discharged 90 d before index admission, N (%)	2390 (29.7)	1335 (33.2)	1055 (26.2)	<0.0001
Year of discharge, N (%)				<0.0001
2004	805 (10.0)	411 (10.2)	394 (9.8)	
2005	844 (10.5)	464 (11.5)	380 (9.4)	
2006	878 (10.9)	496 (12.3)	382 (9.5)	
2007	1048 (13.0)	574 (14.3)	474 (11.8)	
2008	951 (11.8)	507 (12.6)	444 (11.0)	
2009	1037 (12.9)	494 (12.3)	543 (13.5)	
2010	1047 (13.0)	482 (12.0)	565 (14.0)	
2011	788 (9.8)	321 (8.0)	467 (11.6)	
2012	656 (8.1)	278 (6.9)	378 (9.4)	
Length-of-stay (days), median (IQR)	3.3 (2.1–5.8)	3.1 (2.0–5.1)	3.6 (2.1–6.2)	<0.0001
Urgent or emergent admission, N (%)				<0.0001
No	955 (11.9)	318 (7.9)	637 (15.8)	
Yes	7099 (88.1)	3709 (92.1)	3390 (84.2)	
Yes	1395 (17.3)	717 (17.8)	678 (16.8)	
No	6659 (82.7)	3310 (82.2)	3349 (83.2)	
Blood transfusion given, N (%)				<0.0001
Yes	885 (11.0)	319 (7.9)	566 (14.1)	
No	7169 (89.0)	3708 (92.1)	3461 (85.9)	
Parenteral or enteral nutrition, N (%)				<0.0001
Yes	180 (2.2)	43 (1.1)	137 (3.4)	
No	7874 (97.8)	3984 (98.9)	3890 (96.6)	
Discharge status of index admission, ^a^ N (%)				0.0024
Home	4909 (61.0)	2475 (61.5)	2434 (60.4)	
Home with nursing care	1550 (19.2)	786 (19.5)	764 (19.0)	
Sub-acute facility	1363 (16.9)	628 (15.6)	735 (18.3)	
Against medical advice	190 (2.4)	121 (3.0)	69 (1.7)	
Discharge 1 year prior to index admission, N (%)				<0.0001
Home	2852 (35.4)	1562 (38.8)	1290 (32.0)	
Home with nursing care	923 (11.5)	473 (11.7)	450 (11.2)	
Sub-acute facility	774 (9.6)	383 (9.5)	391 (9.7)	
Against medical advice	133 (1.7)	90 (2.2)	43 (1.1)	
No discharge recorded	3372 (41.9)	1519 (37.7)	1853 (46.0)	
Body mass index, N (%)				<0.0001
<18.5 kg/m^2^	182 (2.3)	113 (2.8)	69 (1.7)	
18.5–24.9 kg/m^2^	1587 (19.7)	971 (24.1)	616 (15.3)	
25.0–29.9 kg/m^2^	2223 (27.6)	1082 (26.9)	1141 (28.3)	
≥30.0 kg/m^2^	4062 (50.4)	1861 (46.2)	2201 (54.7)	
Depression or psychosis ever, N (%)	2438 (30.3)	1358 (33.7)	1080 (26.8)	0.0002
Gastroparesis ever, N (%)	683 (8.5)	596 (14.8)	87 (2.2)	<0.0001
Pancreatitis ever, N (%)	410 (5.1)	246 (6.1)	164 (4.1)	0.037
Hypertension ever, N (%)	5630 (69.9)	2631 (65.3)	2999 (74.5)	<0.0001
COPD or asthma ever, N (%)	1551 (19.3)	625 (15.5)	926 (23.0)	<0.0001
Cardiac dysrhythmias ever, N (%)	1431 (17.8)	492 (12.2)	939 (23.3)	<0.0001
Malignant neoplasm ever, N (%)	596 (7.4)	140 (3.5)	456 (11.3)	<0.0001
Drug abuse, N (%)				<0.0001
Never	6262 (77.8)	2990 (74.2)	3272 (81.3)	
History	1403 (17.4)	786 (19.5)	617 (15.3)	
Current	389 (4.8)	251 (6.2)	138 (3.4)	
Current complication of device, graft, or implant, N (%)				<0.0001
Yes	208 (2.6)	53 (1.3)	155 (3.8)	
Current fluid or electrolyte disorder, N (%)	1695 (21.0)	969 (24.1)	726 (18.0)	<0.0001
Charlson comorbidity index, N (%)				<0.0001
0	1271 (15.8)	1269 (31.5)	2 (0.0)	
1–2	2211 (27.5)	914 (22.7)	1297 (32.2)	
3–4	1503 (18.7)	508 (12.6)	995 (24.7)	
5–6	791 (9.8)	370 (9.2)	421 (10.5)	
>6	2278 (28.3)	966 (24.0)	1312 (32.6)	
Outpatient visit, N (%)				<0.0001
Yes	3683 (45.7)	1820 (45.2)	1863 (46.3)	
No	2303 (28.6)	1239 (30.8)	1064 (26.4)	
Unknown	2068 (25.7)	968 (24.0)	1100 (27.3)	

^a^ “Other” category not shown; ^b^ See text for SI units; ^c^ Retinopathy, neuropathy, nephropathy; ^d^ Coronary artery disease, heart failure, stroke, peripheral vascular disease; ACE-i = Angiotensin-converting enzyme inhibitor; ARB = Angiotensinogen receptor blocker; COPD = Chronic Obstructive Pulmonary Disease; DKA = Diabetic ketoacidosis; Ever = current or prior; IQR = Interquartile range; HHS = Hyperglycemic Hyperosmolar Syndrome; No = not recorded.

## Data Availability

Data are available on request due to institutional privacy restrictions on data use.

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
