# Peer review of "Hospital Readmission Risk and Risk Factors of People with a Primary or Secondary Discharge Diagnosis of Diabetes"

_jcm, 2023, doi:10.3390/jcm12041274_

Round 1

Reviewer 1 Report

The present paper aimed to identify risk factors for hospital re-admission in diabetic patients, with a focus on primary discharge diagnosis and secondary ones. This manuscript presents a large amount of data and the analysis that was performed is impressive. I have only some little advice for authors.

The background is well written and provides all relevant aspects to introduce the aim of this study. However, the definition of primary and secondary discharge diagnoses that is depicted in the introduction section is not so clear in the abstract. I would suggest researchers to improve the first part of the abstract adding the aforementioned definition. This would probably help readers to immediately understand the topic of your work.

Materials and methods are properly described and adequate to answer the research question. Statistical analysis is clearly reported.

Results represent the main section of this manuscript. Since authors performed a very deep analysis of several clinical and demographical variables, tables and figures are rich and extensive. Compared to the written explanation of results, tables, particularly Table 1, are too long. Although quite all the reported variables (44/49) showed a statistically significant difference among the two groups of diabetic patients, I would suggest authors to underline in the text the most relevant ones. Moreover, also for Tables 2 and 3 a forrest plot should be considered. This way, a panel of 4 figures (4 plots) would be a solution to help the reader and would offer authors more space to improve the description of results. Again, authors could focus the attention of the reader on the most relevant significant results, that could be deepened then in the Discussion section. C-statistics instead could be included in the supplementary material. In fact, from a clinical and pharmacological point of view, a clinician or a pharmacologist would probably be more interested in aspects related to predictors of hospital readmission than in those related to a correlation analysis. The results of C-statistics should be still reported in the main text.

Discussion is a little bit short and could be improved. A discussion like the one made for “inpatient diabetes team consultation”, for instance, could be done also for other variables. Not for all variables that were analysed, of course. But those that authors believe the most relevant ones, that were described previously in the Results section as suggested before. Strengths and limitations are well described.

Conclusions are supported by results.

Author Response

Comment: The background is well written and provides all relevant aspects to introduce the aim of this study. However, the definition of primary and secondary discharge diagnoses that is depicted in the introduction section is not so clear in the abstract. I would suggest researchers to improve the first part of the abstract adding the aforementioned definition.

Response: Definition was added to the abstract.

Comment: Results represent the main section of this manuscript. Since authors performed a very deep analysis of several clinical and demographical variables, tables and figures are rich and extensive. Compared to the written explanation of results, tables, particularly Table 1, are too long. Although quite all the reported variables (44/49) showed a statistically significant difference among the two groups of diabetic patients, I would suggest authors to underline in the text the most relevant ones. Moreover, also for Tables 2 and 3 a forrest plot should be considered. This way, a panel of 4 figures (4 plots) would be a solution to help the reader and would offer authors more space to improve the description of results.

Response: The tables were condensed and shortened by removing non-significant rows (which are noted in the text), condensing each binary variable into a single row, and deleting “other” categories.

Tables 2 and 3 were replaced with forrest plots. For simplicity, they were labelled individually. We defer to the journal editors on presenting all 4 forrest plots as a panel.

Comment: Discussion is a little bit short and could be improved. A discussion like the one made for “inpatient diabetes team consultation”, for instance, could be done also for other variables.

Response: Commentary on other variables was added to the discussion.

Reviewer 2 Report

The authors have attempted to describe the variables that determine the readmission to hospital within 30 days of patients in a primary or secondary relationship to diabetes mellitus. 

This is an important issue particularly with respect to resource allocation and the growing burden of diabetes mellitus to health economies around the world. 

The paper is well written and clearly presented. The authors conclude from this retrospective study that patients admitted with a primary cause related to diabetes were at higher risk of being readmitted. 

Major concerns are (1) the dataset is more than a decade old and there have been significant changes in therapies and the classifiication of diabetes not captured in the study (2) the relevance of these findings in the post COVID-19 pandemic world were not addressed (3) the key variable of measurement of exposure to hyperglycaemia, haemoglobin A1C was not presented and is a major omission (4) the nature of the readmissions in relation to whether the original treatment was medical or surgical was not presented and (5) an important variable which could be great significance to general diabetes care was the inpatient referral - its shape and form alluded to but not described. The latter is a major weakness of the paper given that care during the admission is a key determinant of the ongoing needs of the patient and makes it very difficult to generalise the findings. 

Th authors have not tested a hypothesis. This is a secondary analysis of the existing data in an electronic patient record. The authors do not explain why and how they would a priori test for a partial overlap of the risk factors in the two groups. The discussion over speculates on some of the associations.

The authors do not postulate on the reasons for the unique factors in each group. It would useful to discuss why for example, in the primary group the uninsured were not at more risk of readmission when co-related socioeconomic factors were determinants.

There is no mortality data reported which may have a bearing on readmission rates being lower in the secondary care group and should have been discussed.

Minor concerns

The tables are unduly long and difficult to follow and could have been presented more succinctly. The categorisation of the levels of blood glucose are not clear and values should also be given in SI units. Categories of mild and severe hypoglycaemia should have been included in the analysis.

Author Response

Comment: Major concerns are (1) the dataset is more than a decade old and there have been significant changes in therapies and the classification of diabetes not captured in the study (2) the relevance of these findings in the post COVID-19 pandemic world were not addressed (3) the key variable of measurement of exposure to hyperglycaemia, haemoglobin A1C was not presented and is a major omission (4) the nature of the readmissions in relation to whether the original treatment was medical or surgical was not presented and (5) an important variable which could be great significance to general diabetes care was the inpatient referral - its shape and form alluded to but not described.

Response: Although the data are old, we believe the findings remain relevant because the risk factors have not changed substantially. We acknowledged in the discussion that lack of diabetes classification and A1C values are limitations.

Relevance of these findings post-COVID-19 was added to the discussion and copied here:

It is difficult to speculate about how readmission risk factors among people with diabetes may have changed since the appearance of COVID-19. Given that the pandemic exacerbated health disparities,26 it is possible that the associations related to access to care, such as outpatient visits, employment status, and insurance were strengthened. While several studies have identified diabetes as a risk factor for readmission among people hospitalized for COVID-19,27 we are unaware of any studies that have examined COVID-19 as a risk factor for readmission among people with diabetes.

Unfortunately, with the data available we are not able to classify original treatment as medical or surgical. This has been added to the discussion of limitations.

The nature of inpatient referral to the diabetes management team was described in the Methods.

Comment: The authors have not tested a hypothesis. This is a secondary analysis of the existing data in an electronic patient record. The authors do not explain why and how they would a priori test for a partial overlap of the risk factors in the two groups. The discussion over speculates on some of the associations.

Response: We agree there is no formal hypothesis testing of partial overlap in risk factors. Therefore, mention of hypothesis testing has been removed from the Intro.

It is unclear to us which associations were believed to have over speculation, so no revision was made in response to this comment.

Comment: The authors do not postulate on the reasons for the unique factors in each group. It would useful to discuss why for example, in the primary group the uninsured were not at more risk of readmission when co-related socioeconomic factors were determinants.

Response: Lacking health insurance was inversely associated with readmission risk in both groups of patients.  Employment status and insurance remained in the multivariable models of both groups; thus, they were each independently associated with readmission and probably capture different domains of risk. We are unaware of other studies that included lack of insurance along with types of insurance that might offer competing evidence to support or refute an expectation that lack of insurance would be associated with a greater risk of readmission.

Comment: There is no mortality data reported which may have a bearing on readmission rates being lower in the secondary care group and should have been discussed.

Response: The lack of mortality data was added to the limitations.

Minor concerns

The tables are unduly long and difficult to follow and could have been presented more succinctly. The categorisation of the levels of blood glucose are not clear and values should also be given in SI units. Categories of mild and severe hypoglycaemia should have been included in the analysis.

Response: The tables were condensed and shortened by removing non-significant rows, deleting “No” rows of binary variables, and deleting “unknown or other”.

Categorization of blood glucose was clarified in the Methods as follows:

Most extreme blood glucose level was based on capillary point-of-care or venous values during the entire hospitalization. The most extreme value was placed into one of three categories: 70-180 mg/dL (3.9-10 mmol/L), 40-69 or 181-300 mg/dL (2.2-3.8 or 10.1-16.7 mmol/L), or <40 or >300 mg/dL (<2.2 or >16.7 mmol/L).

SI units were not added to the table because doing so would have added more rows. Readers are directed to the text to see the SI units with a footnote. Mild and severe hypoglycemia are included in these categories.

Round 2

Reviewer 2 Report

The authors have addressed presentational issues and referred to comments made but have not added any new data. The lack of HbA1c evaluation and details of the inpatient management decisions remain as major omissions. 

Author Response

Unfortunately, there were too many patients missing an A1C value to include A1C in the analysis. We added lack of inpatient management data as a limitation.